# What are housekeeping genes?

**Chintan J. Joshi** [1], **Wenfan Ke** [2], **Anna Drangowska-Way** [2], **Eyleen J. O'Rourke** [2]*,
**Nathan E. Lewis** [1,3,4,5]*

**1** Department of Pediatrics, University of California, San Diego, School of Medicine, La Jolla, California, United States of America, **2** Department of Biology and Cell Biology, University of Virginia, Charlottesville, Virginia, United States of America, **3** Novo Nordisk Foundation Center for Biosustainability at the University of California, San Diego, School of Medicine, La Jolla, California, United States of America, **4** Department of Bioengineering, University of California, San Diego, La Jolla, California, United States of America, **5** National Biologics Facility, Technical University of Denmark, Kongens Lyngby, Denmark

* ejo8b@virginia.edu (EJOR); nlewisres@ucsd.edu (NEL)

**Data Availability Statement:** All relevant data are within the manuscript and its Supporting Information files.

**Funding:** This work was supported by the NIGMS (grant no. R35 GM119850, (to NEL)), Novo

## Abstract

The concept of "housekeeping gene" has been used for four decades but remains loosely defined. Housekeeping genes are commonly described as "essential for cellular existence regardless of their specific function in the tissue or organism", and "stably expressed irrespective of tissue type, developmental stage, cell cycle state, or external signal". However, experimental support for the tenet that gene essentiality is linked to stable expression across cell types, conditions, and organisms has been limited. Here we use genome-scale functional genomic screens together with bulk and single-cell sequencing technologies to test this link and optimize a quantitative and experimentally validated definition of housekeeping gene. Using the optimized definition, we identify, characterize, and provide as resources, housekeeping gene lists extracted from several human datasets, and 10 other animal species that include primates, chicken, and *C. elegans*. We find that stably expressed genes are not necessarily essential, and that the individual genes that are essential and stably expressed can considerably differ across organisms; yet the pathways enriched among these genes are conserved. Further, the level of conservation of housekeeping genes across the analyzed organisms captures their taxonomic groups, showing evolutionary relevance for our definition. Therefore, we present a quantitative and experimentally supported definition of housekeeping genes that can contribute to better understanding of their unique biological and evolutionary characteristics.

## Author summary

Housekeeping genes have often been associated with four main biological criteria—stability in expression across samples, essentiality, participation in cellular maintenance, and evolutionarily conserved. However, the relationship between these criteria has not been much discussed. Housekeeping genes are of broad interest for translational and basic science research. Thus, it is important to present experimental bases for these criteria. In this article, using the Gini coefficient, we compare stability of expression with the other three criteria using 15 transcriptomes spanning 11 organisms. We found that: (1) Previously

Nordisk Foundation (NNF10CC1016517, NNF20SA0066621, to NEL), a Lilly Innovation Fellows Award to CJJ, and funding from the Keck Foundation (EJO'R). This work was also supported by the Novo Nordisk Foundation through the Technical University of Denmark (NNF20SA0066621 to N.E.L.). The funders had no role in study design, data collection and analysis, decision to publish, or preparation of the manuscript.

**Competing interests:** The authors have declared that no competing interests exist.

identified housekeeping genes in humans had lower Gini coefficients regardless of whether they were calculated using tissue or cancer transcriptomes. (2) Genes with low Gini coefficients for human tissues and cancer cells were enriched for significantly different cellular functions. (3) Essential genes for humans, CHO cells, and *C. elegans* were likely to have lower Gini coefficients compared to other genes. (4) Gini coefficients of orthologs conserved across different organisms were able to recapitulate organism-specific information. All these results provide a quantitative view of housekeeping genes across organisms and hint at their evolutionary importance.

## Introduction

Housekeeping genes are often defined as being stably expressed in all cells and conditions [1], essential [2], belong to cellular maintenance pathways [1,3–6] and be conserved [7,8]. The concept of housekeeping genes has aided applied and theoretical biology including the study of evolution. At the grand level, housekeeping genes can be defined as the minimal set of genes required to sustain life [9]. At the practical level, they can be defined as genes stably expressed in all cells of an organism irrespective of tissue type, developmental stage, cell cycle state, or external signal, or as markers of an organism's healthy biological state [2]. At the evolutionary level, they may allow us to define species and higher taxa-specific genomic features [7,10,11] and gene functions [7,8,12] that may drive conservation or change. Thus, knowledge of housekeeping genes can significantly contribute to explorative, basic, and translational studies. However, despite the fundamental and translational utility of the concept of housekeeping gene, for over four decades their definition has remained axiomatic.

However, expression stability is often tested in a handful of cell types and conditions, and then inferred to be stable across most cell types and conditions. Further, expression stability (similar expression across cell types and conditions), function (e.g., belonging to cellular maintenance), essentiality (loss-of-function is lethal), and conservation (in this context, stably expressed and essential across taxa) are four very different properties of a gene (**Fig 1**). While there has been considerable interest in understanding expression stability and essentiality, both these properties have rarely been directly studied within the same experiment for organisms other than humans. Further, the systemic identification of cellular maintenance functions and gene conservation has remained sparse and ambiguous. Further, no study has yet formally tested the relationships or potential linkage between these four properties. For example, it is unknown if or how the expression pattern of a gene informs its essentiality or conservation. Thus, experimental support for the implied association between the properties defining housekeeping genes remains to be reported.

Analysis of large-scale 'omics' data is now commonplace [13], applied to a gamut of questions [14–17] and organisms [18–25]. Further, the availability of single-cell transcriptomics and functional genomics studies for several species is rife with opportunities to study biological principles within and across organisms [23,26,27], including developing a more objective and experimentally supported definition of housekeeping gene. Further, predefining housekeeping genes for an organism would bring several potential benefits. At the experimental level, it can save in troubleshooting for the identification and validation of mRNA expression controls in difficult cell types and unique samples (e.g., seasonal wild organisms or patient biopsies) analyzed via transcriptomics [6] and quantitative real-time PCR (qRT-PCR) [28], and it can provide more robust ways to normalize the growing number of single-cell RNAseq studies.

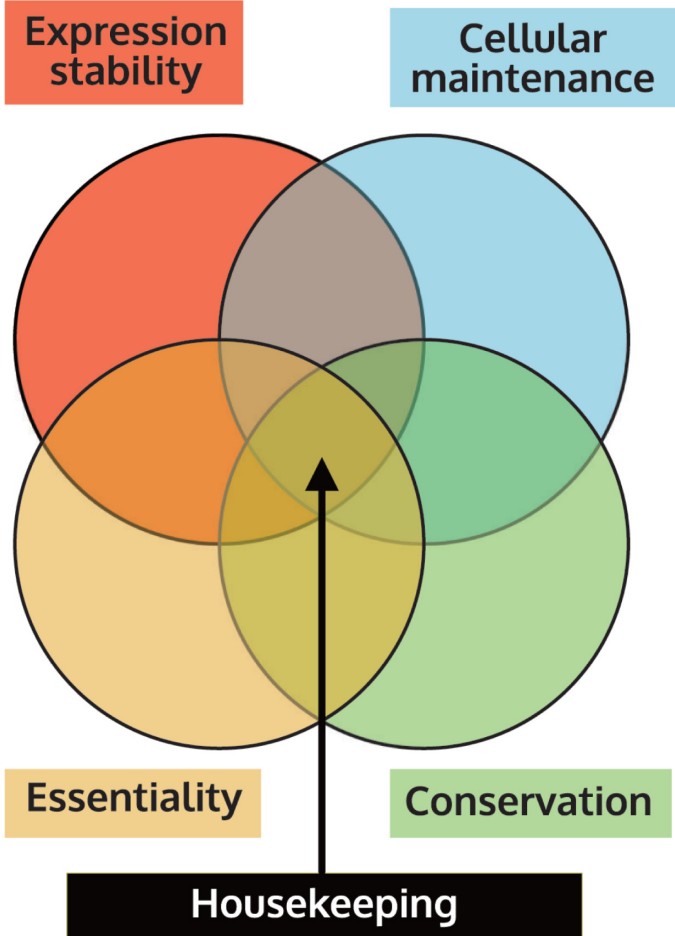

**Fig 1. The relationship between the axiomatically assigned four properties of housekeeping genes remains unexplored.**

Gene essentiality is often described as the loss of cell viability upon deletion of a gene [29,30]. With growing interest in functional genomics studies, CRISPR essentiality screens are becoming increasingly common. While there are plenty of prokaryotes with essentiality screens, only a few mammalian organisms have available datasets which include humans (cancer cells), CHO and mouse [31]. At the more fundamental level, it is important to define the minimum number of genes necessary to sustain life and if and how these genes or their characteristics change across taxa; thus, highlighting the most evolutionarily resilient genes whose functions continue to persevere after millions of years of evolution.

Recent efforts have been made to quantitatively define housekeeping genes [1,28,32,33]. A particularly powerful approach is a mathematical framework called GeneGini [12,32,33], which leverages the Gini coefficient ($G_C$)–a statistical metric quantifying inequality among groups [34]. $G_C$ varies from 0 to 1. In economics, lower Gini coefficients mean lower income inequality. Similarly, in the framework of GeneGini, the $G_C$ of a gene is proportional to the inequality in its expression across samples [32]. Therefore, genes with a low $G_C$ (referred to as Gini genes here on) are stably expressed and could be considered housekeeping genes according to the property of stability. However, many questions remain about Gini genes. Which are the cellular functions carried out by Gini genes? Are Gini genes essential? Do Gini genes retain

their housekeeping status across species? Answers to these questions are central to validate or reject the hypothesis that all four properties–expression stability, basic cellular function, essentiality, and conservation–are linked, and hence should be used to define housekeeping genes.

Here, we used the $G_C$ approach to identify stably expressed genes (Gini genes) across human tissues and cell lines, and across cell types of lower organisms studied using single-cell transcriptomics [19,22,35–38]. We show that, indeed, $G_C$ values are highly correlated across human datasets, supporting the existence of a subset of genes stably expressed across cell types and conditions. Then, we compare the properties of stability ($G_C$ coefficient) to the property of essentiality, as defined by *in vitro* or *in vivo* functional genomics studies. Published CRISPR-Cas9 essentiality screens of Chinese-hamster ovary cells (CHO) [39] and human cell lines [40–42], and new whole-animal essentiality RNAi screening of *C. elegans* done as part of this study, show that essential genes tend to have lower $G_C$, suggesting that stably expressed genes are more likely to be essential. Nevertheless, not all stably expressed genes are essential. Further, genes that are stably expressed and essential are associated with GO terms enriched in basic cellular functions, and also are more likely to be conserved. Therefore, although the four properties informally used to define housekeeping genes are more likely than not to associate with each other, they are not strictly linked. Thus, our analysis provides an experimentally quantitative definition of housekeeping gene, and it establishes a foundation to start asking the most interesting questions including which is the minimum set of genes necessary to sustain life, and if and how these genes change across taxa.

## Results

### $G_C$-defined housekeeping genes are stably expressed across samples of the same species

To quantify $G_C$ across genes and test whether the $G_C$ captures the first property of housekeeping genes (i.e., stable expression across cell types and conditions), we calculated the $G_C$ for all genes in the Genotype-Tissue Expression (GTEx) [36] and Human Protein Atlas (HPA) [22] tissue datasets, and all genes in the NCI-60 Cancer datasets from CellMiner and Klijn et al [37]. We used the 3688 housekeeping genes published by Eisenberg and Levanon [1] as our benchmark gene set. A total of 15687 genes were present in HPA and GTEx, 16052 genes were present in both NCI-60 cancer datasets, and 14327 genes were present in all 4 datasets. From each dataset, we extracted the 3688 genes that had the lowest $G_C$. This gene set allows us to account for the different distribution shapes and number of genes in each dataset (S1A Fig).

Observations that followed: 1) the Gini genes obtained from combining HPA and GTEx (human tissues) covered 81.4% of the 3688 Eisenberg and Levanon housekeeping genes and 2) the NCI-60 cancer datasets covered 69.8% of the 3688 Eisenberg and Levanon housekeeping genes (**Fig 2A**). Lower accuracy in the human cancer datasets relative to the human tissue datasets was expected because the Eisenberg and Levanon housekeeping genes were defined using healthy human tissue transcriptomics [1]. Importantly, the $G_C$ for genes present in any given pair of datasets were highly correlated, and more so for datasets of the same sample types (tissue or cancer samples) (**Fig 2B**). Further, we found that for all datasets, Eisenberg and Levanon's housekeeping genes had significantly low 90th percentile $G_C$ values in each of the datasets compared to the 90th percentile $G_C$ of these datasets (90th percentile ratios: 0.49 for HPA, 0.37 for GTEx, 0.61 for CellMiner, and 0.37 for Klijn et al [37].; S1B Fig), and cross comparison of $G_C$s revealed that the median $G_C$ of these genes was very similar across samples (S2 Fig).

Another common but untested notion is that housekeeping genes are robustly expressed. Although the Gc is informative regarding consistency of expression across samples, it does not

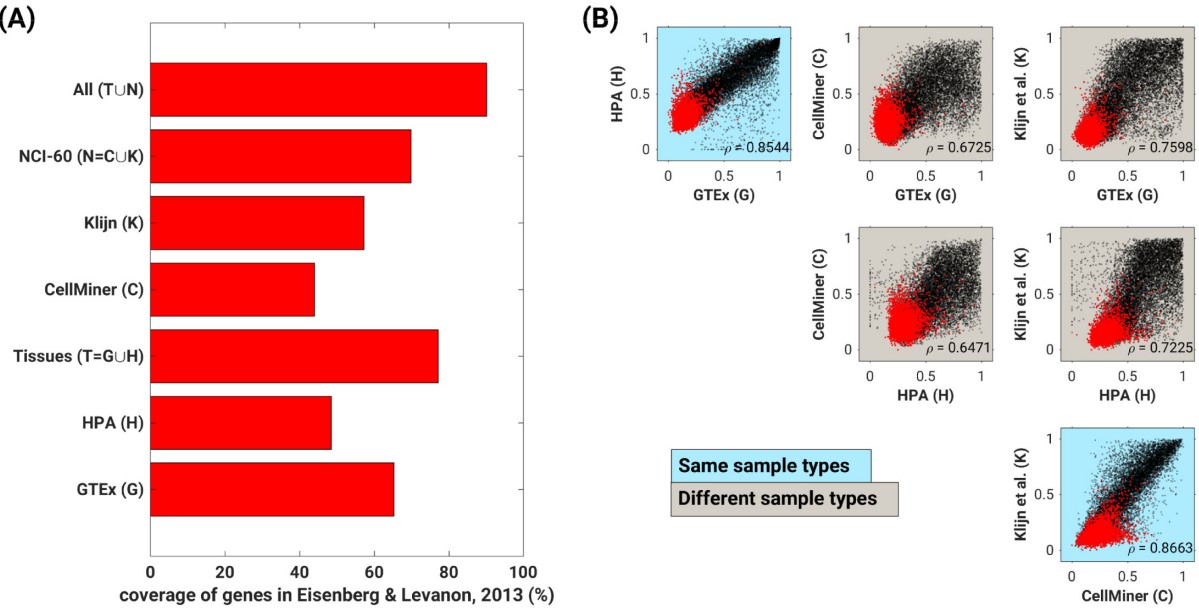

**Fig 2. Gini coefficient captures the first property of housekeeping genes: stable expression. (A)** Coverage of the 3688 Eisenberg and Levanon housekeeping genes [1] among the 3688 genes with lowest Gini coefficients within each dataset. **(B)** Gini coefficients of individual genes are highly correlated across human datasets regardless of sample type; however, spearman correlations within sample types are tighter. Dots represent unique genes. Red specifically depicts housekeeping genes identified by Eisenberg and Levanon [1].

inform about the actual level of expression of genes. To investigate whether housekeeping genes are represented across the range of expression or limited to one or other expression extreme, we used the GTEx [36] and HPA datasets [22] to calculate the median expression levels of the 3688 housekeeping genes previously identified by Eisenberg and Levanon [1]. The expression of these 3688 housekeeping genes varied significantly, from below 10 TPM to over 500 TPMs (**Fig 3A and 3B**). Specifically, the median expression value within the 25th, 50th and 75th percentiles of the HPA dataset, were 12.2, 22.1, and 41.5 transcripts per million (TPM), respectively; whereas within the GTEx dataset, the 25th, 50th and 75th percentile values were 14.2, 24.7, and 46.4 TPM, respectively. Thus, housekeeping genes do not strictly have high expression values.

In summary, genes with low $G_C$ in one sample type are the most likely to have low $G_{Cs}$ in other sample types or datasets, suggesting that $G_C$ captures the property of expression stability, at least within a given species. Additionally, distinct from the current axiom, we found that the majority of the housekeeping genes are expressed at relatively low levels.

## $G_C$-defined housekeeping genes are enriched in cellular maintenance functions

The second property commonly associated to the concept of housekeeping gene is that these genes are involved in basic cellular maintenance functions [1,3–6]. To test this axiom, we performed GO term enrichment analysis on the 3688 genes with lowest $G_C$ identified for each dataset as described above.

First, to define the baseline GO terms, we extracted all GO terms enriched in the complete HPA, GTEx, CellMiner and Klijn *et al* [37] samples. We found 1189 different GO terms that were enriched in at least one dataset (S3 Fig, S1 Table). To minimize the confounding effect of

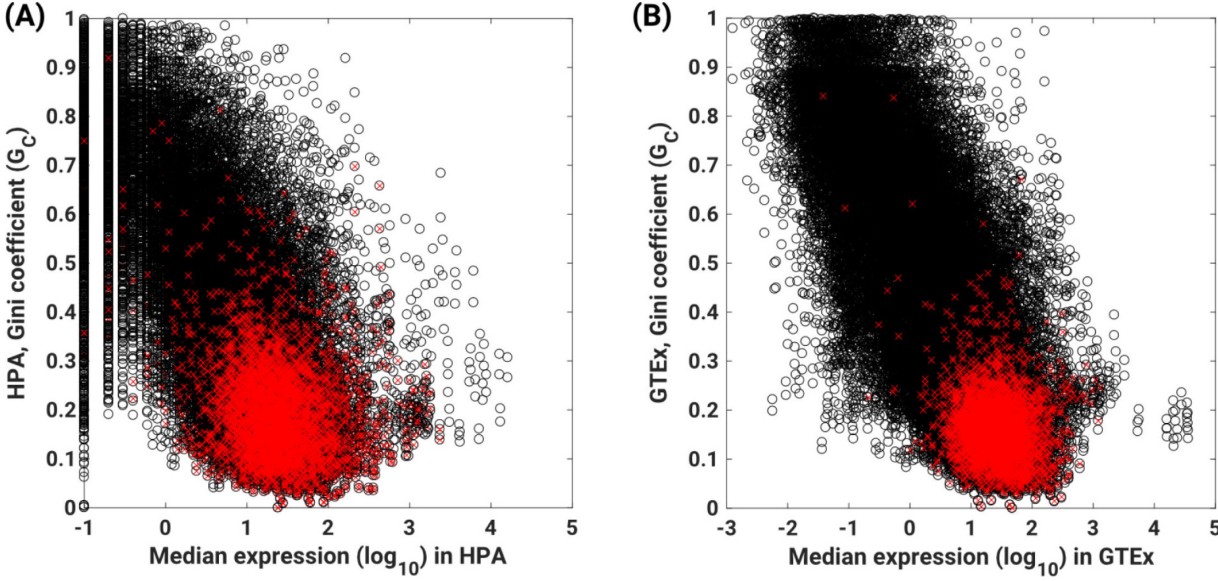

**Fig 3. Housekeeping genes are mostly expressed at relatively low levels. (A-B)** Comparison of median expression levels and Gini coefficients calculated for the **(C)** HPA and (B) GTEx datasets. The spearman correlations between Gini coefficients and median expression values were -0.6545 (p = 0) and -0.7244 (p = 0) for HPA and GTEx, respectively. Dots represent unique genes. Red specifically depicts housekeeping genes identified by Eisenberg and Levanon [1].

using different numbers of subject or query genes for the hypergeometric test, we defined as background frequency of a GO term the representation of the compounded 1189 GO terms among all genes in each dataset. We found that the representation of the 1189 GO terms was highly correlated across datasets ($\rho_{mean}$ = 0.93 across 6 pairs of datasets). Then, we performed the enrichment analysis of the 3688 genes with lowest $G_C$ identified in each dataset (healthy tissue and cancer cell line datasets); details about hypergeometric test can be found in the Methods section.

We found 121 GO terms commonly enriched in all four datasets. The top GO terms with highest coverage were functions classically considered basic cellular maintenance including cell cycle, regulation of mRNA stability, protein folding, protein stabilization, and protein transport, among others. By contrast, the GO terms that were not enriched were related to stress or other context-specific responses including positive regulation of viral life cycle, response to mitochondrial depolarization, antifungal humoral response, and cellular response to misfolded protein, among others. The nature of the GO terms suggests that the GO terms most enriched among genes with low $G_C$ relate to essential growth functions whereas the least enriched terms relate to regeneration and adaptation to stress rather than generation of new cells.

We also found sample type-specific GO terms; for instance, a GO term that was enriched in healthy tissue data but not in cancer cells or vice-versa. We found 77 GO terms enriched exclusively in tissue datasets. The top 10 tissue-specific terms included positive regulation of mitochondrial translation and calcium ion transmembrane transport, branched-chain amino acid (BCAA) catabolic process, 7-methylguanosine mRNA capping, and NIK/NF-κB signaling. Excitingly, mitochondrial translation has been identified as diminished in various cancers [43,44] and BCAA have been identified as essential nutrients for cancer growth [45]. Further, we unveil enrichment of NF-κB across cell types whilst its potential targets are cell-specifically expressed, suggesting an underappreciated strategy for cells to put into action cell-specific

programs driven by common upstream controllers. Thus, at least 3 out of 4 of the GO terms we find enriched in healthy tissues but not enriched in cancer cells correspond to biological functions known to be impaired or dysregulated in several cancers, suggesting that analyses of sample-specific genes with low $G_C$ could contribute to identify disease-specific genes that could become targets to treat or cure these diseases. Beyond its applicability, our list reinforces the notion that housekeeping genes are enriched in cellular maintenance functions.

We also found 70 GO terms enriched only in the cancer datasets, suggesting that cancer cells have housekeeping genes which are different from healthy tissues. These 70 cancer-only GO terms included mitotic cell cycle and replication fork machinery, GO terms associated with the generation of new cells. Further, the overall–as opposed to exclusive–coverage of GO terms enriched in cancer datasets was higher than the overall coverage of GO terms enriched in tissue datasets (S4 Fig), supporting the notion that the oncogenic state is an overall gain of function.

In summary, the data show that low $G_C$ is associated with cell maintenance or cell generation activities, supporting the notion that housekeeping genes defined using the $G_C$ would predominantly be involved in basic cellular functions. Beyond, the $G_C$ identifies housekeeping genes as core genes fundamental to cellular maintenance.

## Housekeeping genes are essential

Third property assigned to housekeeping genes is essentiality–loss of cell viability due to deletion of a gene [29,30]. However, this has only been validated for a handful of genes circumstantially found to be essential. Thus, we test whether essential genes are also stably expressed (low $G_C$) using publicly available functional-genomics studies in mammalian cells (cancer and CHO CRISPR screens), as well as an *in vivo C. elegans* essentiality screen performed specifically for this study.

Essentiality in cancer cell line datasets was extracted from Depmap [40,41] as a CRISPR guide-RNA score (log-fold change of guide-RNA). For essentiality scores in CHO cells, we used a published functional genomics screen that identified 338 essential genes [39] (S5 Table). The accession IDs of transcriptomic data for CHO cells are listed in S6 [46] Table. We found that the 2796 genes that were essential in all 20 cancer cell lines had lower $G_C$ compared to all the genes combined, when calculated using transcriptomics data for the same 20 cell lines from Klijn et al. [37] and CellMiner (**Fig 4A and 4B**). The same was true for the 338 essential genes reported for CHO cells (**Fig 4C**). To facilitate comparison of GO terms between essential genes and genes with low $G_C$ values, the 2796 genes in cancer cells and the 338 genes in CHO cells with the lowest $G_C$ values were chosen. This analysis of cancer cell lines revealed that coverage of GO terms for the essential genes correlated with GO-term coverage in a same number of the lowest $G_C$ genes identified in the Klijn et al. [37] (**Fig 4D**, yellow; ρ = 0.8907) and CellMiner (**Fig 4D**, blue; ρ = 0.8557) datasets. A similar comparison between CHO essential genes and genes with low $G_C$ also resulted in a high correlation (ρ = 0.7055) between low $G_C$ and essentiality in these hamster cells (**Fig 4D**, green). Together these results suggest that Gini genes and essential genes show the same distribution, and hence, are likely largely overlapping.

Gene essentiality is: 1) health-status dependent, 2) context-dependent, and 3) subject to buffering from higher-level cell-cell interactions. Thus, since cell lines studied here are immortalized cultured cells devoid of their physiological and multicellular context, we investigated the correlation between $G_C$ and gene essentiality in a healthy living animal model.

We defined the $G_C$ for *C. elegans* genes using a published whole-body single-cell transcriptomics dataset [18]. We obtained essentiality scores from our *in vivo* RNAi screen for all 1535

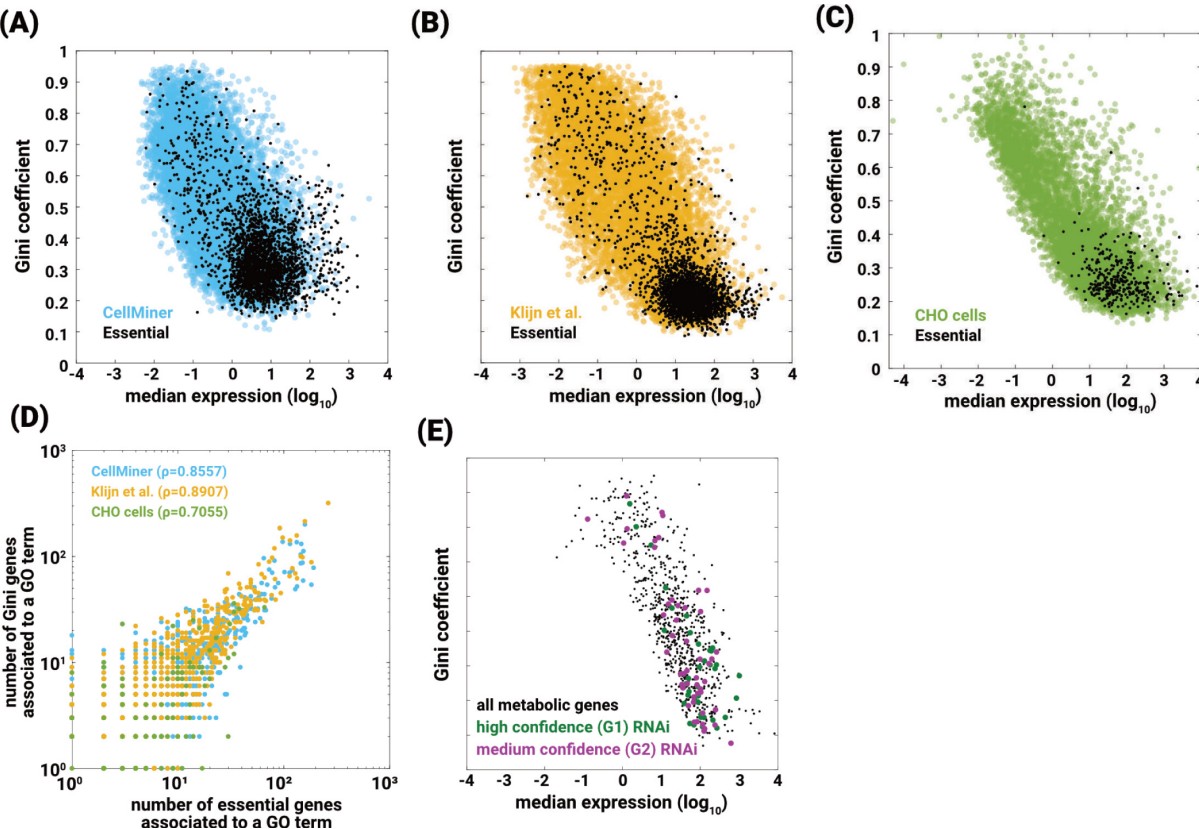

**Fig 4. Gini genes are essential.** Gini coefficients of essential genes (black dots) compared to the housekeeping genes from Eisenberg and Levanon when using (A) CellMiner (Wilcoxon p-value = 8.373 x $10^{-38}$; KS p-value = 2.48 x $10^{-26}$) and (B) Klijn et al. [37] (Wilcoxon p-value = 5.96 x $10^{-18}$; KS p-value = 4.44 x $10^{-23}$) cancer datasets, and (C) CHO datasets. The 2800 genes essential in the 20 cancer cell lines were extracted from DepMap [40,41], and the 338 CHO essential genes were extracted from Xiong et al. [39] (D) GO term coverage of essential genes and that of Gini genes from CellMiner (blue, 0.8557), Klijn et al. [37] (yellow, 0.8907), and CHO (green, 0.7055) are correlated. The left-tailed Wilcoxon p-values when comparing essential genes with the overall datasets are 2.35 x $10^{-146}$ for CellMiner and 9.16 x $10^{-313}$ for Klijn et al. [37] This suggests that essential genes and housekeeping genes are coming from populations with equal medians and similar cumulative distribution functions. The slightly lower correlation in CHO cells is likely due to the fewer number of essential genes reported in the CHO screen. (E) High- and medium-confidence essential genes in healthy *C. elegans* have significantly lower $G_C$ than non-essential genes (p = 4.53 x $10^{-5}$).

predicted metabolic genes in the worm (S4 Table). A gene was considered essential in *C. elegans* if after rearing hatchlings on dsRNA-delivering *E. coli* for 5 days at 25˚C (*E. coli* is the standard lab food source for *C. elegans*) animals were arrested at a pre-adulthood stage (control WT animals are gravid adults at this time). Three relevant phenotypic classes were observed in RNAi-treated *C. elegans*: 1) High-confidence essential, RNAi-treated animals arrested in ≥5 out of the 6 independent RNAi treatments against that gene; 2) Medium-confidence essential, RNAi-treated animals arrested in 3 or 4 out of the 6 independent RNAi treatments; and 3) Wild-type; see Supplementary Methods for further details. We performed a threshold-free statistical test to compare the GC values of different essentiality groups to rest of the genes (for e.g. G1 vs G2 + G3 + G4 + G5). We found that, similar to human and hamster diseased cell lines, high- and medium-confidence essential genes in healthy *C. elegans* had significantly lower $G_C$ than the non-essential genes in the tested pool (**Table 1**, S4 Table).

In summary, the data show that low $G_C$ is associated with essentiality. Inferring a similar association in organisms beyond these two mammalian species and a nematode, we

**Table 1. *C. elegans* essential genes have significantly lower $G_C$ than non-essential genes.**

| Geneset | Geneset definition | Number of genes in the class | Number of genes with $G_C > 0$ [a] | Sign test (median $G_C^{class}$ < median $G_C^{all}$) |
|---|---|---|---|---|
| G1 | High-confidence essential | 48 | 38 | $5.808 \times 10^{-5}$ |
| G2 | Medium-confidence essential | 64 | 49 | 0.0427 |
| G1 + G2 | | 112 | 87 | $4.5304 \times 10^{-5}$ |
| G3 | Wild-type | 1095 | 532 | 0.9814 |
| G4 | Unknown[b] | 174 | 97 | 0.0335 |
| G5 | Untested[c] | 94 | 58 | 0.347 |

(a) Numbers in this column are smaller than in column C because genes with $G_C$ equal to zero were excluded from the analysis.

(b) Unknown includes genes for which, we hypothesize due to strong effects on health and/or development, we were not able to generate large enough populations of worms for quantitative analyses. This hypothesis, and the observations that led us to propose it, are in agreement with the low $G_C$−essentiality correlation p value observed for this class.

(c) Untested corresponds to core metabolic genes that were not tested due to lack of RNAi clone/s or other technical limitations.

hypothesize that the property of essentiality may be conserved across taxa for housekeeping genes defined using the $G_C$ due to the strong selective pressure on essential genes.

## Housekeeping genes preserve organism-specific information

One way to test whether the properties of housekeeping genes defined by having low $G_C$ are conserved across evolution is to compare whether the genes with low $G_C$ in one species significantly overlap with the low $G_C$ genes in another species. Importantly, conservation is the last of the four properties commonly assigned to housekeeping genes; yet not systematically tested.

For our analyses, here, we defined Gini genes as bottom 20th percentile of genes with lowest $G_C$. Thus, we analyzed our low $G_C$ genes in the context of a previously published transcriptomics study of nine organisms [35] that include chicken, platypus, opossum, mouse, macaque, orangutan, gorilla, chimpanzee, and humans. Since most of these organisms do not have a Gene Ontology available, we analyzed the 1:1 orthologs across all these organisms, which corresponded to 5423 genes. Another advantage of limiting the comparison to the 5423 genes is that it removes the bias in the statistical tests otherwise introduced by the sample size.

Different from the analyses of human tissue and cancer-cell datasets, we found that the correlations among the nine species were lower, and the Spearman correlations of the $G_C$ of the 1:1 orthologs distinctively clustered (**Fig 5A and 5B**). The analyses included genes which were identified as Gini gene in at least one of the nine organisms. Gini coefficients accurately clustered non-primate mammals distinctively from primates; thus, reinforcing the value of Gini genes not only to identify conserved housekeeping genes but also distinctive biology.

Principal Component Analysis (PCA) of $G_C$ for 1:1 orthologs was able to reproduce the cluster consisting of primates and the cluster consisting of other organisms (**Fig 5C**, dendrogram) using the across taxa correlations among Gini genes. The first two principal components accounted for 45.4% of the explained variation (S5 Fig). The first principal component separated the primate cluster from the remaining species. Our clustering also revealed top candidate genes responsible (based on their $G_C$ values) for such a clustering. The genes with stable expression across all species (**Fig 5D**, green), primates only (**Fig 5D**, pink), or non-primates only (**Fig 5D**, blue). Interestingly, despite the taxa-specific biology, the coverage of GO terms (S2 Table) associated with the Gini genes was highly correlated across all the organisms (**Fig 5E** (black box)). Together, the results show that individual Gini genes contain important information about species-specific biology, yet higher-level features–such as GO terms–are shared by Gini genes across evolution.

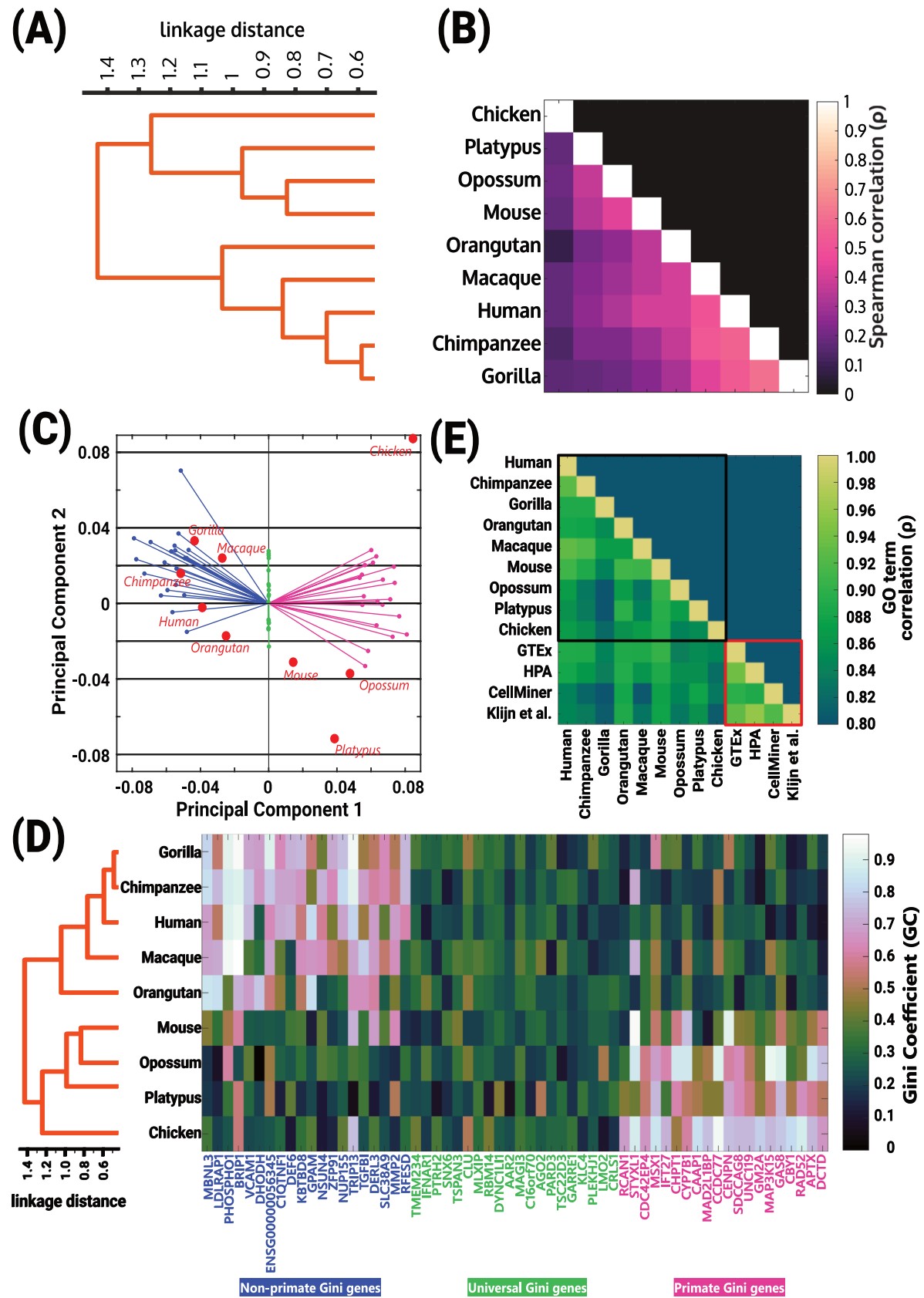

**Fig 5. Gini coefficients accurately capture organism-specific differences.** (A-B) Spearman correlations of Gini coefficients of Gini genes (bottom 20$^{th}$ percentile of $G_C$ in at least one organism) identified using organism-specific transcriptomes capture cluster containing primates. The number of Gini genes with 1:1 orthologs in all organisms is shown using the bar plot on the right of the dendrogram. Please see S10 Fig. for p-values. (C) Principal component 1 (PC1) also captures the cluster containing primates. Also shown are the top 20 primate Gini genes (pink), the top 20 shared Gini genes (green), and top 20 Gini genes in all non-primates (blue) using the principal component coefficients of the first principal components. (D) Correlation among Gini coefficients across different organisms reproduce cluster containing primates (left panel). The Gini coefficients of genes belonging to top 20, middle 20, and bottom 20 coefficients of PC1 are shown (right panel). Left 20 Gini genes are specific to non-primates, middle 20 Gini genes are shared, and right 20 Gini genes are specific to primates. (E) GO term coverage is highly correlated across different datasets, also shown are the GO term correlations with human datasets used in S6 Fig.

## Discussion

Historically, housekeeping genes have been defined as genes that are consistently expressed across tissues, essential, carrying out cellular maintenance, and conserved across species. Here, we used the Gini coefficient ($G_C$) to analyze the concept of housekeeping genes, and found statistical evidence for their existence. Thus, in this only study of its kind, we used $G_C$ as a statistical metric to identify housekeeping genes in humans and numerous other species, allowing us to test suggested criteria of "housekeepingness". The concept has practical importance since genes qualified as "housekeeping" genes have been extensively used for benchmarking and normalizing gene expression results in diverse experimental settings, including qRT-PCR, bulk and single-cell transcriptomics, *in situ* hybridization, western blots, FACS, etc. This has led to common use of genes such as Glyceraldehyde 3-phosphate dehydrogenase (GAPDH). Interestingly, our analyses of GAPDH found large differences in its $G_C$ values across different datasets (S7 Fig), demonstrating the need for a more objective definition of housekeeping gene. While we focus here on the Gini coefficient, other metrics could be used. In fact, a recent study compared 9 different metrics and found that Tau and Gini coefficient had the highest correlation in their respective values across human tissue samples from different datasets [12].

The analyses presented here is threshold free, unless we presented the list of housekeeping genes (for e.g., **Fig 5E**). When listing housekeeping genes, there is the need for specifying thresholds [12]. However, there is no known biological basis for choosing a $G_C$ threshold. Due to lack of biological basis for $G_C$ threshold, biologically meaningful information may be eliminated. Indeed, our PCA of genes showed that the genes with the highest variation in expression (39.5%) did not enable the clustering of the 9 organisms analyzed here (S8 Fig), suggesting that many of the eliminated genes may have critical functions conserved across species. Here, we also show that even though fewer GO terms were enriched in all the datasets or organisms, the coverage of GO terms that were enriched in each dataset was highly correlated across datasets. These results are in line with the previously suggested notion [47] that housekeeping functions, rather than a list of genes, better describe the health state of the organism. Thus, the challenges in quantitatively identifying a set of housekeeping genes confound interpretation and reconciliation of biological functions associated with such genes. Yet, our statistical analyses presented throughout the manuscript suggests that there is value in the notion of "housekeepingness". Devising a framework to identify housekeeping functions will better acknowledge genetic differences amongst organisms and alleviate noisy interpretation of genetic data. Nevertheless, our comprehensive analyses yielded experimentally supported lists of Gini genes (genes with low $G_C$) accompanied by their GO identifiers for different species (S3 Table), which we anticipate will facilitate mechanistic study of housekeeping functions in the future.

Housekeeping genes are often discussed in the context of organisms and not the environmental condition of the organism. For instance, when defining organismal housekeeping genes, the stability of the growth conditions isn't accounted for or discussed. The $G_C$ value for

a gene depends on its pattern and magnitude of expression across diverse time points, cell types, ages, stressors, etc. For instance, we found that when the number of samples is increased, the mean $G_C$ value of the genes increases (S9 Fig). Our GO term analyses clearly showed the significant differences in housekeeping genes in cancer cells versus those for tissues (S4 Fig). While enumerating all the possible environments an organism and its cells will encounter is out of reach, increasing the number of different cell line and tissue samples may yield more stable and accurate Gini genes. And yet, there will remain a possibility of encountering genes that weren't previously considered as housekeeping.

In addition to being stably expressed, housekeeping genes have been axiomatically defined as involved in basic cellular functions and essential. Here we use experimental data to test these notions and show that indeed Gini genes are enriched in GO terms related to cellular maintenance, and are more likely to be essential. Interestingly, for cancer cells, neither the essential genes nor the enriched GO terms fully overlap with those of healthy tissues of the same species suggesting the definition of housekeeping gene proposed here can help distinguish health from disease status, and possibly aid in the identification of targets for the treatment of disease. Specifically, our comparison of GO enrichment between healthy human tissues and cancer cell lines highlighted the shift from regeneration of cells in healthy tissue to generation of new cells in cancer cells. Further, our analyses found that coverage of GO terms that were distinctly enriched in only tissue datasets was lower than those for cancer datasets (S4 Fig). Thus, rather than cancer cells undergoing dysregulation of certain housekeeping genes, our analysis suggests that cancer cells also add distinctive sets of housekeeping genes that differ from the one in healthy tissues. These results were consistent with previous studies that cancers cells and healthy cells do not have the same housekeeping genes; although, some genes do overlap [48]. Further, these "cancer-enriched" housekeeping genes are associated to cellular division or cellular growth. This contrasts with the notion that housekeeping genes belong to cellular maintenance. Our analyses also suggest that cellular maintenance (non-growth needs of the cell) and the growth requirements for cellular division are complementary to one another; and their distinction may not lie in the consistency of expression of a gene.

Another axiomatically accepted property of housekeeping genes is that they are "housekeeping" across species. Our analyses using multi-organism datasets showed high GO term correlation across organisms, suggesting conservation of housekeeping pathways, but no significant conservation of individual housekeeping genes. Further, even for housekeeping pathways, the correlation across species is slightly reduced when compared to the correlation across human datasets (Fig 5E). It is possible that this reduced correlation is due to data limitations. The pathway analyses presented here were done using human GO terms; therefore, genes from any given organism were mapped to the corresponding human orthologs and gene identifiers. In fact, the 1:1 ortholog-based GO term analysis used here resulted in eliminating, on average, 69% of the Gini genes from each of the 9 organisms analyzed. Therefore, availability of gene ontology beyond model organisms can provide molecular insight into species-specific biology. Nevertheless, Gc was able to capture taxa-specific biology.

The analyses of worms presented weaker p-values compared to the other datasets (Fig 4E, Table 1). This could be due to various reasons: (i) the environmental conditions for transcriptomics (L2) and RNAi were different (embryonic to adult), (ii) the stability of L2 worms, and (ii) the only single cell RNA-seq (scRNA-seq) dataset in our analyses. The worms used for scRNA-seq were L2 [18], a developmental stage, and could have different housekeeping genes than L3/L4. As we observed for humans (tissues and cancer cells), the differences in environmental conditions could potentially lead to different housekeeping genes. As discussed earlier, the stability of the environmental conditions isn't usually accounted for when defining housekeeping genes. Finally, while the scRNA-seq offers spatial resolution, i.e. expression of genes in

individual cells, some genes with very low expression at cellular level will not be captured [49,50]. This could, in part, explain the weaker correlations in worms compared to the analyses for other organisms. While scRNA-seq technologies is a rapidly growing interest, it remains to be seen if housekeeping genes can be reconciled by integrating bulk and scRNAseq.

Key molecular similarities likely underlie the physiological similarities between related species. By crossing Gini coefficients with CRISPR-Cas9 essentiality screens and GO terms we may have captured some of these key molecular similarities as our analysis was able to distinguish primate from non-primates. On the other hand, even though animals seem phenotypically very different they share molecular similarities that we can capture at the level of GO terms, even if not at the level of specific gene IDs. Nevertheless, what is essential across environmental contexts and taxonomic groups, if anything, is worth future investigation.

In conclusion, the current definition of housekeeping genes invokes four criteria–stability, maintenance, essentiality, and conservation. It does not address exceptions when a gene has a slightly higher $G_C$ value than the threshold, or if it meets only two or three of these criteria. For housekeeping genes to exist, there needs to be a reconciliation of plasticity in organismal process when encountering changing environmental conditions and rigidity in its definition. Further, during the span of evolution, the environmental conditions changed regardless of whether these conditions were stable for the organism. More work is needed on molecular understanding of housekeeping genes through the course of evolution. Indeed, in part, these questions may be addressed by either discarding the concept of housekeeping gene or recognizing the "housekeepingness" of a gene as a more continuous metric. Given the influence of environment on the list of organismal housekeeping genes, additional criteria maybe needed for growth conditions for samples. Our study only scratches the surface of the answer to practical and fundamental questions and shows the need for organism-specific tools and models; but not just for model organisms, we need models for a diverse set of organisms. Our study suggests that analysis of the ever-increasing "omics" datasets presents an opportunity for better understanding of the biological functions fundamental to sustain life and drive evolution.

## Method

### Literature search

We performed a literature search using Harzing's Publish or Perish 7 [51] to extract the top 1000 hits from Google Scholar for the query keywords: housekeeping, genes, maintenance, and required. The list of top 1000 papers was downloaded to an excel sheet for further analysis and visualization on MATLAB.

### Data extraction

Transcriptomic datasets were obtained from various sources (Table 2). To resolve differences in gene identifiers, we mapped all to NCBI Entrez gene identifiers using BioMart, within the Ensembl website. When genes did not map to an NCBI gene identifier, we discarded these genes from the analyses. All the transcriptomics datasets are available from the Github repository.

### Gini coefficient ($G_C$)

The $G_C$ measures the inequality in frequency distribution of a given parameter (e.g., levels of income, income mobility [54], education [55], etc.) compared to the frequency distribution of total population [34]. For analysis of transcriptomic data, the parameter is expression of a given gene and is compared against the total gene expression is distributed across different

**Table 2. Data sources used for this study.**

| Organism (sample type) | Data source | Modifications |
|---|---|---|
| **Housekeeping genes** | | |
| Humans | Eisenberg and Levanon [1] | Gene nomenclature within the original dataset was changed to NCBI Entrez gene identifiers. Due to issues related to mapping gene nomenclature, the original set of 3804 was reduced to 3688 genes. |
| **Transcriptomics** | | |
| Human (tissues) | HPA [22] | - |
| | Brawand et al., 2011 [35] | Converted to TPM from read per base |
| | GTEx [36] | |
| Human (NCI-60 cancer cell lines) | CellMiner [52] | - |
| | Klijn et al. 2015 [37] | - |
| *C. elegans* (cell types) | Cao et al., 2017 [18] | - |
| Chicken, Platypus, Orangutan, Bonobo, Gorilla, Chimpanzee, Macaque, Mouse, Opossum (tissues) | Brawand et al., 2011 [35] | Converted to TPM from read per base using the following formula: $TPM_i^s = \left[ \left( \frac{r_i^s}{L_i^s} \right) / \sum \left( \frac{r_i^s}{L_i^s} \right) \right] \times 10^6$ $r_i^s$ = read (gene i, sample s) $L_i^e$ = exon length (gene i) |
| Chinese hamster ovaries (cell lines) | See S6 Table for accession IDs; Shamie et al. 2021 [46] | |
| **Essentiality screens** | | |
| Human (NCI-60 cancer cell lines)–CRISPR-Cas9 | DepMap [40–42] | - |
| CHO cell lines–CRISPR-Cas9 | Xiong et al. 2020 [39] and S5 Table | - |
| *C. elegans* (cell types)—RNAi | New data provided here by Eyleen J. O'Rourke; method described in Ke et al. 2018 [53] and Supplementary text. | - |

samples [32]. The $G_C$ is calculated as the ratio of area between the Lorenz curve and line of equality over the total area under the line of equality. The Lorenz curve is the graphical representation of the distribution of a given parameter; and is given by Eq (1):

$$L(F(x)) = \frac{\int_{-\infty}^{x} t\, f(t)dt}{\mu} \qquad 1$$

where $\mu$ denotes the average, *f(x)* denotes the probability density function, and *F(x)* denotes the cumulative distribution function. The calculation was implemented in MATLAB (R2016b), for which the code is available at GitHub (https://github.com/LewisLabUCSD/gene-gini-matlab).

## Gene Ontology (GO) enrichment

Due to lack of availability of unique gene ontologies for the different organisms discussed in the study, genes of the organisms that mapped to the human ortholog genes were used to identify the respective GO term. Here, hypergeometric tests were used to check whether the number of genes associated to a GO term, in the query list, are more significant given the distribution among GO terms in the subject gene list. GO terms associated to human genes were downloaded from Gene Ontology Consortium webpage (http://current.geneontology.org/products/pages/downloads.html). All analysis was focused only on the Biological Process (P) aspect. All p-values were calculated using hypergeometric test for overrepresentation reported after correction using the Benjamini Hochberg FDR. A GO-term was considered enriched if the corrected p-values were below 0.05.

## Supporting information

**S1 Text. Supplementary text contains results showing that the literature survey for definition of housekeeping genes and analyses for Glyceraldehyde 3-phosphate dehydrogenase not being a good candidate as a housekeeping gene for every dataset.** Supplementary method in this text is method used for scoring whole-body *C. elegans* RNAi screens. Also shown are the supplementary figures, captions, and captions for the supplementary tables.
(DOCX)

**S1 Fig. List of housekeeping genes published by Eisenberg & Levanon have low Gini coefficients when using GTEx, HPA, CellMiner, or Klijn et al. [37] datasets.** (A) The distribution of the Gini coefficients of all the genes for each of the datasets. (B) The distribution of the Gini coefficients of only the list of genes published by Eisenberg & Levanon.
(TIF)

**S2 Fig. Cross comparison of Gini genes across datasets.** Gini genes identified using one dataset had lower Gini coefficient in the other dataset. The grey background plots are those where the P-value calculated suing Wilcoxon rank sum test was not significant (p <0.00001).
(TIF)

**S3 Fig. Volcano plots for GO term enrichment analysis for GTEx (A), HPA (B), CellMiner (C), and Klijn et al [37]. (D).** The colors green, red, and grey indicate the GO terms which were over-represented, under-represented, and not enriched. The size of the bubble indicates number of genes belonging to a GO term. The x-axis represents the ratio of number of hits in the list of Gini genes to those achieved at random.
(TIF)

**S4 Fig. Coverage of GO terms enriched in only cancer datasets was higher than coverage of GO terms enriched in only tissue datasets.** Red bars for cancer cells and Black bars are for tissues. The full data used to prepare this plot is in S1 Table. Included in this plot are 70 GO terms which are only enriched in cancer datasets but in neither of the tissue datasets, and 77 GO terms which are only enriched in tissue datasets but in neither of the cancer datasets.
(TIF)

**S5 Fig. Percentage of explained variation in Principal Component Analysis.** The PCA was performed using 1:1 ortholog Gini coefficients which were calculated using transcriptomes in Brawand et al. [35] to cluster organisms.
(TIF)

**S6 Fig. GO term coverage is highly correlated across human datasets.** HPA/GTEx comparison represented in blue, CellMiner/GTEx in green, and Klijn [37]/GTEx in red.
(TIF)

**S7 Fig. Glyceraldhyde 3-phosphate dehydrogenase (GAPDH) may not be a good choice for housekeeping gene.** Gini coefficients were converted to percentiles (x-axis) using each of the datasets (y-axis). GAPDH has high Gini coefficient in most of the datasets.
(TIF)

**S8 Fig. A scatter plot showing principal components 1 & 2 if genes and organisms switched places in the analysis.** The first principal component which captures majority of explained variation does not explain Gini values in either of the organisms as all of them lie to the right-hand side of the plot.
(TIF)

**S9 Fig. Increasing number of samples increases the mean $G_C$ values of all genes.** Each dot represents the mean $G_C$ value of all the genes. The figure was generating by randomly selecting arbitrary number of samples (less than the total number of samples) 100 times.
(TIF)

**S10 Fig. Comparison of Gini coefficients of 9 different organisms using Spearman correlations.** (A) Spearman correlations of Gini coefficients of Gini genes (bottom $20^{th}$ percentile of $G_C$ in at least one organism) identified using organism-specific transcriptomes capture cluster containing primates.
(TIF)

**S1 Table. GO term enrichment analysis of Gini genes identified using human tissues and cancer cell line datasets.** Shown are 1244 terms enriched in at least one of the following human datasets–GTEx, HPA, CellMiner, Klijn et al. [37]
(XLSX)

**S2 Table. GO term enrichment analysis of Gini genes identified using transcriptomes published in Brawand et al [35].** Shown are enriched terms in each of the organisms. The $G_C$ threshold was set as bottom $20^{th}$ percentile. The results only account for genes which have a 1:1 orthologs in all the organisms.
(XLSX)

**S3 Table. List of Gini genes identified using each of the 15 transcriptomes—GTEx, HPA, CellMiner, Klijn et al. [37], 9 organisms in Brawand et al. [35], *C. elegans* cell types, and CHO cells.** Each tab of the excel file is one transcriptome. For 9 organisms from Brawand et al. [35], 20% of genes with lowest $G_C$ were considered as Gini genes.
(XLSX)

**S4 Table. List of C. elegans genes in each of the 5 groups classified using RNAi screens.**
(XLSX)

**S5 Table. List of essential genes in CHO cells as identified by Kai et al. 2020 [39].**
(XLSX)

**S6 Table. List of accession IDs from where CHO transcriptomics were assembled.**
(XLSX)

## Author Contributions

**Conceptualization:** Chintan J. Joshi, Nathan E. Lewis.

**Data curation:** Chintan J. Joshi, Nathan E. Lewis.

**Formal analysis:** Chintan J. Joshi.

**Funding acquisition:** Chintan J. Joshi, Eyleen J. O'Rourke, Nathan E. Lewis.

**Investigation:** Chintan J. Joshi, Eyleen J. O'Rourke, Nathan E. Lewis.

**Methodology:** Chintan J. Joshi.

**Project administration:** Eyleen J. O'Rourke, Nathan E. Lewis.

**Resources:** Wenfan Ke, Anna Drangowska-Way, Eyleen J. O'Rourke.

**Software:** Chintan J. Joshi.

**Supervision:** Eyleen J. O'Rourke, Nathan E. Lewis.

**Validation:** Chintan J. Joshi.

**Visualization:** Chintan J. Joshi.

**Writing – original draft:** Chintan J. Joshi, Eyleen J. O'Rourke, Nathan E. Lewis.

**Writing – review & editing:** Chintan J. Joshi, Eyleen J. O'Rourke, Nathan E. Lewis.

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
