## [Decision Letter · Decision Letter 0]

9 Mar 2022

Dear Dr. Lewis,

Thank you very much for submitting your manuscript "What are housekeeping genes?" for consideration at PLOS Computational Biology.

As with all papers reviewed by the journal, your manuscript was reviewed by members of the editorial board and by several independent reviewers. In light of the reviews (below this email), we would like to invite the resubmission of a significantly-revised version that takes into account the reviewers' comments. Please in particular pay attention to concerns raised with respect to choice of data sets and gene selection.

We cannot make any decision about publication until we have seen the revised manuscript and your response to the reviewers' comments. Your revised manuscript is also likely to be sent to reviewers for further evaluation.

Sincerely,

Christoph Kaleta

Associate Editor

PLOS Computational Biology

Ville Mustonen

Deputy Editor

PLOS Computational Biology

Reviewer's Responses to Questions

**Comments to the Authors:**

Reviewer #1: The manuscript "What are housekeeping genes?" by Joshi et al. revisits the definition of housekeeping genes. The authors summarize commonly-used criteria for housekeeping gene properties and argue that the inter-relationship between different criteria has not yet been systematically assessed. A quantitative measure of Expression Stability is proposed as a continuous proxy for 'housekeepingness' and its relation to the gene function, essentiality and conservation across taxonomic groups is analyzed on the basis of publicly available data resources.

The manuscript is concise and very-well written and structured. Plots support the main results and statements. I have a few comments/suggestions:

- Lines 103-104: I understand that the authors aimed to have the same number of candidate housekeeping genes (n = 3688) between the Eisenberg & Levanon benchmark gene set and the different test data sets (HPA, GTEx, CellMiner, Klijn). Yet, I would expect that the coverage of Gini value-based selection of housekeeping genes in the test data sets with the benchmark is, in part, dependent on the number of genes in each data set. E.g. What are the total gene numbers in each data set? The first paragraph in the results section states only sizes of combined data sets. A different approach to gene selection could be to select x% of genes with the lowest Gini values per data set and thereby better account for the different gene numbers in the different data sets. Perhaps the authors could elaborate on the motivation behind their gene selection approach and contrast their choice with alternative selection methods.

- One of the mentioned criteria for 'housekeepingness' is that these genes are consistently expressed across tissues and conditions (i.e. Expression Stability). If I understood it correctly, this criterion is directly represented in the calculated Gini coefficient with low values indicating similar expression rates across treatments/tissues. Thus, the interpretation of low Gini coefficients to be associated with consistent expression can only be made for the conditions and tissues tested, while the possibility exists that inferred 'housekeeping' genes could display differential expression in a yet untested condition or e.g. not yet investigated organism developmental stage. This risk/limitation still exists with the housekeeping gene definition proposed by the authors. I invite the authors to elaborate on this common issue in housekeeping gene definition and the limitations of quantitative housekeeping definition that is proposed.

*minor comments*

- Lines 66-68: Since I'd expert a broad readership for this paper, it would be great to provide a brief description why it is important to identify minimal gene sets supporting life.

- Lines 103-104: Also, the sentence should ideally be split into two; the second starting with "This gene set allowed [...]"

- Line 112: "had low G C values". This is a bit vague/imprecise in my opinion. I would suggest to express this statement more quantitatively, e.g. in terms of median Gc values and their position in the Gc value distribution.

- The gene coefficient symbol "G_C" is sometimes written with C in the subscript, sometimes with a lower case "c". This should be unified.

Reviewer #2: The definition and conception of housekeeping genes were ambiguous for decades. This study presented a quantitative approach to define the housekeeping genes from four aspects, including expression stability, cellular maintenance, essentiality and conservation. Expression stability was scored by Gini coefficient (Gc); cellular maintenance was detected by GO term; essentiality was identified through the influence of development levels; conservation was defined by finding the consensus signature through different organisms. The study is timely and may be interest for the readers of the journal.

I have number of comments to improve the presentation of the paper before its publication.

Major comments

1. The potential link among the four terms (expression stability, cellular maintenance, essentiality and conservation) were still unclear, and some data applied in the study may not fit. I suggest the authors discuss this in the manuscript.

2. The application of cancer data may confuse the readers. It had been reported that housekeeping genes could have different performance in cancer compared with normal tissues, and the results in this study also support the conclusion. For example, the low overlap fraction of low Gc genes and housekeeping genes, the enriched GO terms of cancer data had low connection with basic cellular maintenance, and so on. Please discuss this in the manuscript.

3. The application of whole-body single cell RNA-seq data of worm may confuse the reader. Compared with bulk data, the major advantage of scRNA-seq is the expression data at cell type level, but the advantage in the usage of cell type data has not been clearly shown in the paper. The disadvantage of scRNA-seq technique that it has low sensitivity to low expression genes, may have an adverse effect while calculate Gc. Additionally, the sc data was generated from an individual worm larval stage, which can cause unstable results. The authors may discuss this in the manuscript and revise it accordingly.

4. Some results were not clearly described. In figure3, the relationship between the expression level and Gc lack the support of statistics, for example, spearman correlation. In figure4D, the direct evidence of the overlapping between essential genes and low Gc genes was required, which can be showed by Jaccard index, hypergeometric distribution, Fisher exact test, and so on, instead of the number of genes of enriched GO terms. In figure5B and figure 5E, p values were required to show the significance of spearman correlation coefficient.

5. This study processed transcriptomics datasets from different data sources such as HPA and GTEx. How many tissues were analysed by the study? Please describe the details in the method section. In addition, it would be interesting to add discussion about the median expression (in Figure 3A &B) of housekeeping gene (3688 genes published by Eisenberg and Levanon across tissues.

6. The benchmark gene selection. The published paper from Eisenberg and Levanon found 3804 housekeeping genes, and in the presented manuscript, only 3688 genes were adopted. There should be a description about the criteria of benchmark gene selection. The authors may discuss this in the manuscript and revise it accordingly.

7. Why cancer expression profiles (NCI-60) were included when screening housekeeping genes? The results here showed housekeeping gene/GO term overlapping between tissue and cancer were not very consistent, so will this shrink or enlarge the possible range of candidate genes in functional manner? Considering some cancer types have poor 3-year survival rate, which could lead to unstable cell condition when sampling for RNA-seq, so it necessary to also include expression profiles of chronic diseases.

8. GO terms in cancers showed different emphasized aspect differ from normal tissues, which concentrating in cell cycle processes. This is conflicting with the definition of housekeeping genes.

9. The authors suggested housekeeping functions, rather than housekeeping genes, may be suitable in describing the organism status. It’s more convincing if there is comparison between the housekeeping functions and genes when it come to the correlation among different tissues/cell lines.

Minor comments

1. As described in the abstract, except human data, data from other 12 organisms were included. However, only 11 species can be found in the main text. The results of GO terms containing p-values are expected.

2. In figure4, the arrangement of plots should follow the same order.

3. Authors mentioned that data without TPM value were converted from read per base, we suggest to provide the detailed calculation.

4. Figure 5B, the legend shows spearman correlation but the caption is about Jaccard similarity.

5. The authors should state clearly which statistic method they used in correlation calculation part of Figure 2B.

6. The criteria of “low Gc value” should be stated clearly.

Reviewer #3: What are housekeeping genes?

PCOMPBIOL-D-22-00176

Review

Summary of the research and overall impression

The Authors discuss the concept of housekeeping genes analyzing four properties: stable expression across cell types and conditions, involvement in basic cellular maintenance functions, essentiality, conservation across evolution.

The Authors use the GC approach to identify stably expressed genes (Gini genes) across human tissues and cell lines, and across cell types of lower organisms studied using single-cell transcriptomics. Then they compare the properties of stability (GC coefficient) to the property of essentiality, as defined by in vitro or in vivo functional genomics studies.

GC values are highly correlated across human datasets, supporting the existence of a subset of genes stably expressed across cell types and conditions.

Further, genes that are stably expressed and essential are associated with GO terms enriched in basic cellular functions, and also are more likely to be conserved.

Although the four properties informally used to define housekeeping genes are more likely than not to associate with each other, they are not strictly linked. In conclusion the results show the need for organism-specific tools and models.

The topic is of interest for the scientific community and the work presents a paradigm that can be broadly and usefully applied in the field of quantitative gene expression analyses.

Major issues

In general, the methods are adequate, but the manuscript needs to improve in clarity and readability. Please find below my suggestion.

Introduction

The first part of the Introduction is confounding. You should state a general definition of HF (a large class of genes that are constitutively expressed, subjected to low levels of regulation in different conditions and perform biological actions that are fundamental for the basic functions of the cell CIT Eisenberg E and Levanon EY: Human housekeeping genes, revisited. Trends Genet 29: 569-574, 2013.) Then, my suggestion is to follow the general idea of the four criteria: stability, maintenance, essentiality, and conservation, always in the same order along the manuscript. If the order will be maintained, the manuscript will improve in readability.

In particular, in the Introduction I think the state referring to normal conditions (“they can be defined as genes stably expressed in all cells of an organism under normal conditions irrespective of tissue type, developmental stage, cell cycle state, or external signal, or as markers of an organism’s healthy biological state is confounding”) is confounding. My suggestion is to focus on the four properties and to add a paragraph to discuss the idea the normal and pathological condition show different patterns of gene expression and also different HK genes.

Results

The Results session is well organized because it follows the description of the four properties, but since the Authors performed analyses on human tissues, Human NCI-60 cancer cell lines, other organisms you should better specify the results obtained in each tissues/cells/organism in each paragraph of the Result session.

Lines 187-244 Session results, Paragraph “Housekeeping genes are essential”:

I suggest starting the paragraph with a definition of “essentiality”.

Is it possible to obtain data related to essentiality in human normal cell? Why did you focus on essentiality analysis only in cancer cell lines and C. elegans? You should clarify that you have performed analyses only for those two cell types and add a paragraph in the discussion.

Discussion

Lines 286-299 You should add references: not only the references already used in the Introduction, but you should also add references related to previous research (e.g. Eisenberg E and Levanon EY: Human housekeeping genes, revisited. Trends Genet 29: 569-574, 2013; Casadei R, Pelleri MC, Vitale L, Facchin F, Lenzi L, Canaider S, Strippoli P and Frabetti F: Identification of housekeeping genes suitable for gene expression analysis in the zebrafish. Gene Expr Patterns 11: 271-276, 2011; Zhang Y, Li D and Sun B: Do housekeeping genes exist? PLoS One 10: e0123691, 2015; Caracausi M et al. Systematic identification of human housekeeping genes possibly useful as references in gene expression studies, MOLECULAR MEDICINE REPORTS, 2017)

As said before, I suggest the adding of a paragraph discussing the results with a specific reference to different cell types/organisms and a paragraph discussing the choice of analyzing only cancer cell lines and C. elegans in the context of the property essentiality.

Minor issues

Line 174

“could be become targets”: please cancel “be”

**Have the authors made all data and (if applicable) computational code underlying the findings in their manuscript fully available?**

Reviewer #1: Yes

Reviewer #2: Yes

Reviewer #3: Yes

PLOS authors have the option to publish the peer review history of their article (what does this mean?). If published, this will include your full peer review and any attached files.

Reviewer #1: No

Reviewer #2: **Yes: **Adil Mardinoglu

Reviewer #3: No
---

## [Decision Letter · Decision Letter 1]

10 Jun 2022

Dear Dr. Lewis,

We are pleased to inform you that your manuscript 'What are housekeeping genes?' has been provisionally accepted for publication in PLOS Computational Biology.

Best regards,

Christoph Kaleta

Associate Editor

PLOS Computational Biology

Ville Mustonen

Deputy Editor

PLOS Computational Biology

Reviewer's Responses to Questions

**Comments to the Authors:**

Reviewer #2: The authors reviewed the paper based on the reviewer comments and I recommend its publication in its current form.

Reviewer #3: The Authors made several changes and the readability of the manuscript has benne improved

**Have the authors made all data and (if applicable) computational code underlying the findings in their manuscript fully available?**

Reviewer #2: Yes

Reviewer #3: Yes

PLOS authors have the option to publish the peer review history of their article (what does this mean?). If published, this will include your full peer review and any attached files.

Reviewer #2: **Yes: **Adil Mardinoglu

Reviewer #3: No

---

## [Editor Report · Acceptance letter]

29 Jun 2022

PCOMPBIOL-D-22-00176R1 

What are housekeeping genes?

Dear Dr Lewis,

I am pleased to inform you that your manuscript has been formally accepted for publication in PLOS Computational Biology. Your manuscript is now with our production department and you will be notified of the publication date in due course.

With kind regards,

Zsofia Freund
